# Effects of Micronutrients and Heavy Metals on Endothelial Function and Cardiovascular Risk in the Face of Environmental Changes

**DOI:** 10.3390/cimb48010041

**Published:** 2025-12-27

**Authors:** Agata Doligalska-Dolina, Marcin Dolina, Amanda Zoń, Emilia Główczewska-Siedlecka, Karolina Osińska, Gary Andrew Margossian, Carla Liana Margossian, Katarzyna Napiórkowska-Baran

**Affiliations:** 1Chojnice Specialist Hospital, 89-600 Chojnice, Poland; doli285@wp.pl (A.D.-D.); dolina.marcin.p@gmail.com (M.D.); 2Jan Biziel University Hospital No. 2, 85-168 Bydgoszcz, Poland; amandazon6@gmail.com; 3Department of Geriatrics, Faculty of Health Sciences, Collegium Medicum Bydgoszcz, Nicolaus Copernicus University, 87-100 Torun, Poland; egs@cm.umk.pl; 4Student Research Club of Clinical Immunology, Department of Allergology, Clinical Immunology and Internal Diseases, Collegium Medicum Bydgoszcz, Nicolaus Copernicus University, 87-100 Torun, Poland; osinska.karolinax@gmail.com; 5Faculty of Medicine, New York Medical College, Valhalla, NY 10595, USA; garymargossian@gmail.com; 6Jacobs School of Medicine and Biomedical Sciences, University at Buffalo, Buffalo, NY 14203, USA; carlamargossian@gmail.com; 7Department of Allergology, Clinical Immunology and Internal Diseases, Collegium Medicum Bydgoszcz, Nicolaus Copernicus University, 87-100 Torun, Poland

**Keywords:** micronutrients, heavy metals, endothelium, oxidative stress, cardiovascular risk, environment, hypertension, obesity, diabetes

## Abstract

Dynamic environmental changes significantly affect trace element balance and exposure to toxic metals, influencing vascular homeostasis. The endothelium, as a key regulator of vascular tone and inflammation, is highly sensitive to fluctuations in micronutrient and heavy metal concentrations. This review summarizes current evidence on the molecular mechanisms by which essential trace elements, such as zinc, selenium, copper, and magnesium, support endothelial function through antioxidant defense, nitric oxide regulation, and anti-inflammatory signaling. Conversely, exposure to heavy metals including cadmium, lead, mercury, and arsenic induces oxidative stress, disrupts nitric oxide bioavailability, and promotes endothelial dysfunction, accelerating the pathogenesis of many diseases. The paper examines how these alterations contribute to the development of major cardiovascular diseases and outlines preventive measures to reduce associated risks. Understanding these interactions is crucial for society’s health amid growing environmental challenges.

## 1. Introduction

In recent decades, dynamic environmental changes have been observed as a consequence of anthropogenic activity, geological processes, and climate variability. Environmental factors, alongside genetic determinants, exert a significant influence on systemic homeostasis [1,2]. Both the deficiency and excess of trace elements, as well as the presence of heavy metals in the environment, can markedly affect human physiology, particularly in the context of cardiovascular health. Within the spectrum of environmental influences on the cardiovascular system, special attention has been devoted to endothelial function. The vascular endothelium, despite constituting a single layer of cells, plays a crucial role in cardiovascular regulation, and endothelial dysfunction represents one of the key pathophysiological mechanisms underlying the development of circulatory disorders [3]. This article discusses the role of trace elements in modulating vascular endothelial function.

With regard to beneficial effects on the endothelium, the molecular mechanisms of action of micronutrients such as selenium, copper, and zinc have been extensively investigated [4,5]. Conversely, the toxic properties of certain heavy metals, including cadmium, lead, and arsenic, have been studied in relation to endothelial activity [6]. Dynamic environmental factors, such as industrial pollution and climate change, are of particular importance, as they contribute to fluctuations in the concentrations of trace elements in the biosphere [7]. Identifying these interrelationships is essential for understanding current public health challenges and for developing effective preventive strategies.

The aim of this article is to provide a comprehensive analysis of the effects of selected trace elements and heavy metals on vascular endothelial function in the context of increasing environmental burden. The discussion focuses on the contribution of these elements to the pathogenesis of cardiovascular diseases, their potential health implications, and avenues for prevention. This topic remains of particular relevance given the rising prevalence of cardiovascular disorders, which are at least partially associated with environmental toxin exposure and disturbances in trace element homeostasis. The link between these factors and arterial hypertension has been especially emphasized in the literature [8].

This review also examines the molecular and biochemical mechanisms that play a pivotal role in the initiation and progression of endothelial alterations. Particular attention is given to oxidative stress, ionic imbalances, and epigenetic modifications, which, under varying conditions of environmental exposure, influence endothelial integrity [9,10].

## 2. The Vascular Endothelium

To accurately explore the pathomechanisms of the processes described, it is essential to comprehensively discuss the role of the vascular endothelium in cardiovascular physiology at the molecular level. A detailed analysis of its structure, properties, and regulatory mechanisms enables a deeper understanding of its function as both a structural component of the vascular barrier and an active modulator of vascular processes. The complexity of endothelial function highlights how deviations from homeostasis can contribute to the development of cardiovascular diseases. This process becomes particularly relevant in the context of toxic environmental exposures and disturbances in trace element balance.

The vascular endothelium constitutes the fundamental structural unit of the vascular system. Endothelial cells line the interior surface of all blood and lymphatic vessels [11].

### 2.1. The Role of the Endothelium in the Life Cycle of the Vascular System

During early embryonic development, the formation of the vascular network occurs in parallel with hematopoiesis. Endothelial cells originate from the mesoderm, which gives rise to angioblasts and hemangioblasts—common precursors of endothelial and hematopoietic cells, respectively. The vascular system undergoes extensive remodeling and maturation through the recruitment of smooth muscle cells and pericytes. Additionally, hemogenic endothelium is capable of generating hematopoietic stem cells [12]. Hellmut G. Augustin et al. clearly described the stages of the vascular system life cycle as vasculogenesis, angiogenesis, maturation, and senescence.

In the phase of vasculogenesis, endothelial cells form the primary vascular plexus by differentiating from mesenchymal precursor cells. Their development is governed by growth factors such as vascular endothelial growth factor (VEGF), which initiates vessel formation and spatial organization within developing tissues.

During angiogenesis, the endothelium facilitates the expansion of the vascular network by generating new vessels from pre-existing ones. Endothelial cells respond to environmental and metabolic stimuli by activating specific transcriptional programs and migrating along chemotactic gradients. They also coordinate vascular remodeling and the recruitment of supporting cells, such as pericytes, which stabilize the newly formed structures.

In the maturation phase, the endothelium maintains the structural and functional integrity of blood vessels. It serves as a selective barrier, regulates the exchange of molecules between the bloodstream and surrounding tissues, and sustains the balance between pro- and anti-angiogenic factors. Active crosstalk between endothelial cells and surrounding fibroblasts, pericytes, and immune cells enables precise adaptation of local vascular properties to the metabolic needs of tissues.

With advancing age, endothelial senescence occurs, leading to impaired barrier function, reduced responsiveness to angiogenic stimuli, and enhanced pro-inflammatory and prothrombotic activity. The consequent reduction in vascular density limits tissue perfusion and contributes to the pathogenesis of cardiovascular diseases [13].

Differentiation of angioblasts into arterial or venous endothelial cells occurs prior to the formation of distinct vascular structures. Key molecular determinants include the tyrosine kinase receptor EphB4 and its ligand ephrinB2, which define venous and arterial identity, respectively. The interaction between VEGF and its receptor VEGFR2 (also known as KDR/Flk1) activates Notch signaling, inducing ephrinB2 expression in arterial endothelial progenitors. In the absence of Notch activation, endothelial cells default to a venous phenotype [12]. Table 1 and Figure 1 presents the types of endothelial cells in individual blood and lymphatic vessels [14].

The transcription factor chicken ovalbumin upstream promoter–transcription factor II (COUP-TFII), expressed specifically in venous endothelial cells, suppresses Notch signaling, thereby reinforcing venous identity. At a later stage, the transcription factor Prospero homeobox 1 (Prox1) is expressed in a subset of these cells, initiating their differentiation toward the lymphatic endothelial lineage. COUP-TFII cooperates with Prox1, and their reciprocal regulation is crucial for determining lymphatic endothelial cell fate.

Proper vascular network formation requires coordinated angiogenic activity of endothelial cells. During this process, the cells proliferate, migrate, and elongate, forming endothelial cords that subsequently undergo lumen formation (tubulogenesis). This sequence of events is precisely orchestrated by interconnected signaling pathways and transcriptional networks [12].

### 2.2. Organ-Specific Heterogeneity

Endothelial cells (ECs) display substantial structural and functional heterogeneity, which is strictly dependent on their organ-specific localization.

In the brain, ECs form a critical component of the blood–brain barrier (BBB). They are characterized by the absence of fenestrations, the presence of tight intercellular junctions, and low transcytotic activity, all of which ensure high selectivity of permeability. In contrast, the liver contains sinusoidal ECs with numerous fenestrae and loose intercellular junctions, allowing for the passage of macromolecules between blood and hepatic tissue.

In the heart, ECs form a dense and compact structure that reinforces the vascular barrier. They exhibit high metabolic and angiogenic activity and express signaling molecules such as neuregulin-1 and nitric oxide (NO).

Renal ECs possess a highly organized structure supporting the filtration process. Their gene expression profiles and epigenetic features reflect the anatomical and functional specificity of this organ.

In the lungs, ECs participate in both gas exchange and immune regulation, exhibiting expression of proteins such as angiotensin-converting enzyme (ACE) and developmentally regulated molecules.

Tumor-associated endothelial cells, in contrast, display disrupted architecture, excessive permeability, and form irregular, disorganized vessels. Their phenotype is largely determined by the tumor microenvironment.

The growing application of single-cell analytical techniques has significantly advanced the understanding of endothelial heterogeneity, enabling the identification of tissue-specific EC signatures under physiological and pathological conditions [14].

### 2.3. Surface Structures and Intercellular Junctions

Among the endothelial surface structures, particular attention should be given to the glycocalyx, which plays a pivotal role in maintaining vascular homeostasis. The multilayered glycocalyx facilitates molecular interactions while reducing direct contact between ECs and blood components. It is primarily composed of glycoproteins and proteoglycans [15]. Through its intracellular domain, the glycocalyx contributes to mechanotransduction, allowing for detection of shear stress and regulation of vascular wall tension, thereby influencing local hemodynamics. Furthermore, its negative electrostatic charge repels blood cells from the vessel wall, supporting blood fluidity and preventing undesired adhesion [16].

Endothelial cells are interconnected via several types of intercellular junctions, including adherens junctions, tight junctions, and gap junctions, which are organized alternately along the lateral membrane from the apical to the basal pole. These junctions maintain the structural coherence and integrity of the endothelial monolayer.

Key adhesion molecules include E-selectin, P-selectin, and members of the immunoglobulin superfamily, such as intercellular adhesion molecule-1 (ICAM-1), intercellular adhesion molecule-2 (ICAM-2), platelet endothelial cell adhesion molecule-1 (PECAM-1), and vascular cell adhesion molecule-1 (VCAM-1). In addition to their structural role, these junctional complexes are crucial mediators of intracellular signaling related to cell proliferation, apoptosis, and inflammatory activation [17].

### 2.4. Functions of the Vascular Endothelium

As an integral tissue, the vascular endothelium plays a key role in oxygen and nutrient delivery, regulation of blood flow, and modulation of immune cell trafficking [11]. Initially regarded as a passive vascular barrier that permitted bidirectional exchange of gases and solutes between blood and tissues, the endothelium is now recognized as a dynamic endocrine and paracrine organ. ECs actively participate in the regulation of hemostasis, maintaining a delicate balance between procoagulant and anticoagulant states. It is now well established that endothelial cells synthesize and release numerous bioactive mediators, thereby exerting a central role in vascular tone regulation and inflammatory modulation [18]. Among vasodilatory and antiplatelet agents, two of the most critical are nitric oxide (NO) and prostacyclin (PGI_2_) [16].

Nitric oxide (NO) is a gaseous free radical with high diffusibility and potent biological activity. It is synthesized in ECs by endothelial nitric oxide synthase (eNOS) from L-arginine and molecular oxygen, with the participation of cofactors such as nicotinamide adenine dinucleotide phosphate (NADPH), flavin adenine dinucleotide (FAD), flavin mononucleotide (FMN), heme, and tetrahydrobiopterin (BH_4_). Its production depends on calcium-calmodulin activation and post-translational modifications, including phosphorylation of eNOS.

NO plays a fundamental role in maintaining vascular homeostasis. By diffusing into smooth muscle cells, it activates soluble guanylyl cyclase (sGC), leading to increased cyclic guanosine monophosphate (cGMP) levels and subsequent activation of protein kinase G (PKG). The downstream effect is a reduction in intracellular Ca^2+^ concentration, relaxation of vascular smooth muscle, and vasodilation [19,20].

Prostacyclin (PGI_2_), a member of the eicosanoid family, acts synergistically with NO to regulate vascular tone. Its synthesis increases in response to stimuli such as hypoxia, shear stress, and exposure to acetylcholine or serotonin. PGI_2_ induces vasodilation by activating potassium channels and elevating cyclic adenosine monophosphate (cAMP) levels in smooth muscle cells [17].

In parallel, the endothelium also produces vasoconstrictive mediators, including endothelin-1 (ET-1) and angiotensin-converting enzyme (ACE). An imbalance favoring pro-inflammatory, prothrombotic, and vasoconstrictive factors leads to endothelial dysfunction, a key pathogenic event in atherosclerosis and other cardiovascular diseases [16].

### 2.5. Endothelial Dysfunction

Given the multilevel significance of the endothelium, it serves as a central regulatory element of the vascular system. However, numerous environmental and physiological factors can contribute to its damage.

In 1998, Hunt and Jurd identified five principal mechanisms underlying endothelial cell dysfunction: (1) loss of vascular integrity, (2) increased expression of adhesion molecules, (3) acquisition of a prothrombotic phenotype, (4) cytokine production, and (5) upregulation of human leukocyte antigen (HLA) molecules. These are further influenced by the organ-specific heterogeneity of ECs under both physiological and pathological conditions [21]. In endothelial cells, nitric oxide (NO) is a key mediator of vascular homeostasis. NO induces vasodilation and inhibits several pathological processes, including the expression of pro-inflammatory cytokines and chemokines, low-density lipoprotein (LDL) oxidation, vascular smooth muscle cell proliferation, and thrombosis. Reduced NO bioavailability, resulting from decreased NO production and its increased degradation by the superoxide anion, is a major contributor to endothelial cell injury [22].

As described by Hojjat Naderi-Meshkin and Wiwit Ananda Wahyu Setyaningsih, endothelial dysfunction can be divided into early and late phases. The process begins with endothelial activation triggered by stimuli such as oxidative stress, inflammatory cytokines, or mechanical shear forces [23]. This activation stimulates nuclear factor kappa B (NF-κB) signaling, leading to increased expression of genes encoding VCAM-1, ICAM-1, E-selectin (ES), P-selectin (PS), and chemokines such as monocyte chemoattractant protein-1 (MCP-1). These molecules facilitate the adhesion of monocytes and lymphocytes to endothelial cells, resulting in endothelial injury and the release of inflammatory mediators, which further amplify the inflammatory response [24].

The subsequent increase in vascular permeability allows plasma proteins and inflammatory cells to extravasate into the surrounding tissues. Simultaneously, vascular tone regulation becomes impaired, manifesting as abnormal vasodilation and vasoconstriction [23].

Activated endothelial cells express tissue factor (TF), which initiates the extrinsic coagulation pathway. In addition, Weibel–Palade bodies of endothelial cells release, among other mediators, von Willebrand factor (vWF), which promotes platelet adhesion and aggregation and enhances inflammatory processes through leukocyte recruitment and complement activation [24].

As dysfunction progresses, a prothrombotic state develops through the expression of procoagulant factors, accompanied by enhanced oxidative stress, which perpetuates cellular injury. A key pathological transition is the endothelial-to-mesenchymal transition (EndMT), in which ECs acquire mesenchymal characteristics, promoting fibrosis and tissue remodeling.

Chronic stress and damage ultimately lead to endothelial apoptosis and senescence, resulting in microvascular rarefaction, the loss of small vessels that diminishes tissue perfusion and contributes to hypoxia. The cumulative outcome of these processes comprises a spectrum of adverse vascular changes (Figure 2) [23].

Environmental factors also influence endothelial function. Chronic stress is associated with elevated glucocorticoid levels, which increase circulating concentrations of pro-inflammatory cytokines and reduce NO synthesis. Concurrently, enhanced catecholamine production is linked to increased activation of the renin–angiotensin–aldosterone system (RAAS) and elevated lipid peroxidation, thereby promoting inflammatory responses and exacerbating endothelial injury [25]. The described pathophysiological framework emphasizes that the endothelium functions not merely as a passive barrier but as an active regulator of inflammation, coagulation, vascular tone, and cell survival. A comprehensive understanding of the sequential stages of endothelial dysfunction is essential for identifying potential therapeutic targets in cardiovascular and chronic inflammatory diseases [23]. Table 2 presents vasoactive factors with vasoconstrictor and vasodilator effects [26].

## 3. The Role of Trace Elements and Heavy Metals

Under changing environmental conditions, increasing exposure to toxic heavy metals and disturbances in the balance of trace elements may initiate or accelerate the processes described above. Such factors influence oxidative stress, ionic imbalance, and signaling pathways, thereby modulating endothelial functions at multiple levels. Consequently, environmental disruptions of elemental homeostasis may represent an important component in the pathogenesis of endothelial-dependent vascular diseases.

### 3.1. Trace Elements

#### 3.1.1. Zinc

Zinc (Zn) is a trace element involved in numerous cellular pathways, serving as a cofactor in enzymatic systems and modulating the activity of enzymes such as superoxide dismutases and metalloenzymes [4]. It also acts as a signaling molecule. The molecular mechanisms of cardiovascular function depend on zinc signaling pathways [27]. Moreover, twenty-four ZIP transporters that increase cytosolic Zn levels and twelve metallothioneins (MTs) responsible for Zn regulation and ROS neutralization have been identified in human cardiac tissue [27,28,29].

Since Zn serves structural, regulatory, and catalytic roles in many metabolic processes, maintaining its homeostasis is essential. Disruption of Zn homeostasis affects endothelial function as well as genomic and proteomic modifications linked to cardiovascular diseases [28,29]. Zn is a divalent cation, of which only about 0.1% of total body zinc is present in human serum, mostly bound to proteins that ensure their structural stability and enzymatic activity. Intracellular zinc is primarily localized in the cytoplasm and nucleus, with minimal amounts in the cell membrane. Its intracellular distribution occurs through four major pools involved in Zn homeostasis regulation:

Zn bound to metalloproteins/metalloenzymes as a structural component or cofactor;

Zn bound to metallothionein (MT), acting as a receptor and donor during disturbances;

Zn stored in intracellular organelles and vesicles, delivered via Zn transporters;

Free cytoplasmic Zn participating in signal transduction.

The concentration of free cytoplasmic zinc is low, ranging from picomolar to nanomolar [29,30]. Cellular Zn homeostasis is maintained through “zinc buffering” and “zinc muffling.” Buffering ensures a constant level of free Zn by binding it to cytosolic proteins, while muffling operates during transient changes by activating Zn transporters to move ions into subcellular compartments or out of the cell, accompanied by the stimulation of Zn-binding proteins. The metal-responsive transcription factor-1 (MTF-1) regulates the transcription of Zn transporters (ZnT) and MTs in response to Zn fluctuations. This feedback between Zn concentration and transcription factors allows for precise homeostatic control. The ZnT and MT families play crucial roles in cardiovascular disease pathogenesis through their regulation of intracellular Zn balance [27,29].

Intracellular Zn modulates endothelial nitric oxide synthase (eNOS) by promoting the conversion of catalytically inactive monomers to active dimers involved in NO synthesis. The Zn chelator N,N,N′,N′-tetrakis(2-pyridylmethyl)ethylenediamine (TPEN) induces eNOS dimer dissociation, while Zn deficiency reduces eNOS expression and activity. Additionally, NO influences intracellular Zn homeostasis by stimulating MT-mediated Zn release, which in turn enhances eNOS activity, forming a positive feedback loop [29,30].

There is evidence that Zn reduces the expression and activity of inducible NOS (iNOS) in endothelial cells. This is another finding suggesting a beneficial effect of Zn on the cardiovascular system, as iNOS is expressed only under pro-inflammatory conditions. Its detrimental role in disease has been demonstrated in studies where it led to excessive NO synthesis in response to inflammatory signals.

Furthermore, Zn exhibits anti-inflammatory properties in processes implicated in cardiovascular diseases. Endothelial cells exposed to various stimuli secrete chemokines and recruit monocytes to vessel walls, an interaction (monocyte–EC adhesion) that drives atherogenesis. The zinc finger–homeodomain transcription factor ZEB1 suppresses expression of the chemokine CXCL1, which mediates this interaction. Zn deficiency enhances inflammation by upregulating VCAM-1 and ICAM-1 [29]. Low Zn levels also increase pro-inflammatory cytokines such as TNF-α, IL-2, IL-6, and C-reactive protein, while cytokines themselves can alter ZnT expression. Moreover, Zn deficiency impairs protein function, including protein kinase C, and disrupts macrophage and monocyte activity [28,30].

The nuclear transcription factor NF-κB is a central mediator in pro-inflammatory signaling and atherosclerosis progression. Zn supplementation inhibits NF-κB activation in vascular ECs, whereas deficiency enhances its activity [28,29]. Other zinc finger–containing transcription factors, including Krüppel-like factors KLF2, KLF4, and KLF11, also exert anti-inflammatory effects in ECs by suppressing NF-κB signaling. Zn further inhibits NF-κB by modulating peroxisome proliferator–activated receptors (PPARs): Zn deficiency decreases PPARα and PPARγ expression, leading to enhanced NF-κB DNA binding. Increased expression of the NF-κB inhibitory protein A20 has been linked to higher Zn concentrations in ECs.

Zn also plays an antioxidant role by catalyzing and supporting antioxidant enzyme synthesis. Its deficiency increases oxidative stress in ECs, while Zn supplementation restores redox balance.

Zn affects endothelial apoptosis as well. ZEB1 protects the p21 protein from oxidative stress–induced damage, preventing apoptosis and senescence. Zn deficiency promotes LPS-induced apoptosis via caspase-3 activation, whereas Zn supplementation reverses this effect. Studies in human aortic EC cultures demonstrated an inverse correlation between Zn concentration and caspase-3 activity [29,30] (Figure 3).

#### 3.1.2. Selenium

Selenium (Se) is a trace element that enhances cell survival through antioxidant protection, regulation of autophagy, inhibition of inflammation, and prevention of apoptosis by modulating both intrinsic and extrinsic signaling pathways. It also influences the Nrf2, JAK/STAT, and PARP pathways. Low Se intake increases the risk of cardiovascular diseases [5].

The antioxidant action of Se is primarily mediated by selenoproteins that neutralize reactive oxygen species (ROS). Se is incorporated as selenocysteine at the catalytic sites of these proteins [5,31,32,33]. Adequate Se and selenoprotein levels maintain redox balance, ensuring proper endothelial and smooth muscle function [34]. Se supplementation increases glutathione peroxidase (GPx) levels in humans and enhances GPx activity in response to toxic exposures such as cadmium, lead, arsenic, and certain drugs.

Reduced GPx activity is observed in Keshan disease, a cardiomyopathy associated with Se deficiency leading to myocardial fibrosis and degeneration [5,31].

The selenoprotein thioredoxin reductase (TrxR) also exhibits antioxidant properties and protects against endothelial dysfunction. Studies show decreased TrxR levels under Se-deficient conditions, reversed upon Se or oxidant exposure (e.g., HOCl, H_2_O_2_) [5].

The selenoprotein TRxR1 is involved in the reduction of coenzyme Q10 (CoQ10) to its active form, ubiquinol, and selenium deficiency may impair the cells’ ability to achieve optimal CoQ10 levels, which counteract lipid peroxidation. This coenzyme functions as a potent antioxidant and a central electron carrier in the respiratory chain. Meta-analyses have shown that its supplementation positively affects endothelial and cardiac function, and low levels of CoQ10 have been observed in patients with ischemic heart disease and cardiomyopathy [31].

It has also been demonstrated that the three thioredoxin reductases: TRxR1, TRxR2, and TRxR3, participate in cardiovascular function by mitigating oxidative stress in response to pressure overload–induced hypertrophy and left ventricular remodeling [33].

Thus, GPx and TRxR, whose activity is selenium-dependent, modulate oxidative stress and lipid oxidative modification. They also reduce inflammation, support proper vascular reactivity, and limit platelet aggregation. Through these mechanisms, they help regulate cardiovascular function and protect against disease [31].

It has been shown that selenium deficiency destabilizes oxidative stress regulation by inhibiting the Nrf2 pathway, which is responsible for activating the cellular response to oxidative stimuli and preventing cellular damage. Nrf2 is a key regulator of antioxidant defenses within cells [5]. Human cardiomyocyte studies show that Se deficiency impairs mitochondrial function and increases ROS production [33].

Selenium regulates cardiomyocyte apoptosis by modulating the mRNA expression of caspase family genes involved in this process, as well as influencing the activity of several other cell death receptors [5,33]. Activation of these genes induces apoptosis via both death receptor and mitochondrial pathways, with the anti-apoptotic factor BCL-2 serving as the principal regulator. Selenium deficiency is associated with increased mRNA levels of caspase-3 and caspase-9, along with decreased BCL-2 expression, all of which promote cardiomyocyte death [32,33]. Additionally, selenium reduces the expression of FAS and TNF receptor genes involved in the formation of cell death–inducing signaling complexes [5].

There are also reports that selenium deficiency affects the cardiovascular system through modulation of microRNAs (miRNAs). Potential miRNAs involved in this regulation have been identified. In one study, cardiac dysfunction associated with selenium deficiency was linked to five upregulated miRNAs, which influence oxidative stress, cell survival, and inflammation, mainly affecting cell death pathways [33].

Selenium deficiency has been shown to inhibit autophagy in cardiomyocytes by increasing expression of the ATG7 gene, which regulates this process [5].

Cardiomyocyte studies also demonstrated that selenium deficiency reduces mitochondrial STAT3 activity, potassium channel expression, and overall mitochondrial function [5,32]. STAT3 proteins play a protective role in the myocardium following ischemia–reperfusion injury by modulating mitochondrial function, which is crucial for cardioprotection [5]. One study showed that pretreatment with selenium modulated STAT3 phosphorylation in this context [5,32]. The STAT/JAK pathway participates in the inflammatory response, mediating cytokine signaling [32].

Furthermore, selenium reduces PARP activity in scopolamine-treated cells. Inflammation pathogenesis is linked to PARP-1 signaling, as its upregulation can lead to mitochondrial dysfunction and cell death [5] (Figure 4).

#### 3.1.3. Copper

Copper (Cu) is a catalytic cofactor in antioxidant defense, energy metabolism, and mitochondrial respiration [35]. Maintaining Cu homeostasis is critical for the cardiovascular system, as Cu deficiency leads to its sequestration in mitochondria and affects the modulation of this trace element and vascular function. Conversely, Cu excess contributes to oxidative stress. These processes result in endothelial cell (EC) dysfunction and cardiovascular diseases [35,36].

Cardiac function relies on mitochondrial activity and high energy production, which requires adequate Cu levels in cardiac tissue [37]. Cytochrome c oxidase (CCO), essential for cellular respiration, contains Cu in its structure, and mitochondrial respiratory efficiency in the heart depends on intact mitochondrial respiration. Cu deficiency results in reduced CCO production due to decreased COX17 transport to SCO1/SCO2 and COX11. Additionally, expression of mitochondrial-associated proteins, such as the peroxisome proliferator-activated receptor gamma coactivator 1α (PGC-1α), increases; this protein is necessary for mitochondrial biogenesis regulation. Consequently, mitochondrial dysfunction arises, characterized by altered mitochondrial structure and impaired proliferation, contributing to atherosclerosis and heart failure. It has also been shown that fatal cardiovascular diseases are associated with CCO activity and expression, leading to stiffening of cardiomyocyte fibers. Furthermore, cardiac and other tissues exhibit reduced ATP and phosphocreatine levels, along with increased ADP and inorganic phosphate. Ultimately, myocardial damage occurs due to additional processes, including changes in membrane and mitochondrial cristae structures, which induce mitochondrial rupture and impair mitochondrial metabolism [36,38].

Cu is a cofactor of superoxide dismutase (SOD). Its deficiency causes endothelial dysfunction by reducing NO synthesis and increasing superoxide anion levels. These processes decrease SOD activity, culminating in enhanced oxidative stress and reduced vasodilation, contributing to atherosclerosis [37,38]. Another notable protein is ceruloplasmin (CP), which transports over 90% of plasma Cu and affects the activity of Cu-dependent enzymes such as SOD, thereby helping eliminate ROS. CP also participates in iron mobilization as a ferroxidase. Consequently, Cu deficiency reduces CP oxidative activity and may be indirectly linked to cardiac diseases associated with disrupted iron homeostasis [38].

Cu deficiency can also alter cardiac gene expression. Involved processes include increased expression of TNF-α and genes regulating cardiac contractility and calcium handling, which contribute to impaired cardiac function under low-Cu conditions.

It has been demonstrated that leukocyte adhesion to activated ECs can be disrupted by Cu deficiency due to decreased expression of adhesion molecules VCAM-1 and ICAM-1 [37].

Studies show that ischemic injury is associated with reduced cardiac Cu levels, along with decreased hypoxia-inducible factor 1 (HIF-1)-dependent angiogenesis and lower expression of genes involved in angiogenesis and glycolysis. During myocardial infarction, HIF-1 is regulated by its HIF-1α subunit. Cu influences HIF-1 function by stabilizing HIF-1α, binding to hypoxia-responsive element (HRE) sequences of target genes, and forming transcriptional complexes. Cu also stimulates expression of Cu-dependent genes such as VEGF, thereby affecting HIF-1–HRE interactions. Furthermore, Cu modulates lysyl oxidase (LOX) synthesis, involved in vascular maturation via Ras-related C3 botulinum toxin substrate 1 (RAC1), ATOX1, and ATP7A. Cu efflux inhibits these mechanisms, leading to myocardial damage via reduced vascular wall tension, increased myocardial fragility, and impaired angiogenesis. Increased Cu efflux occurs under hypoxia and ischemia [36].

Disrupted Cu ion homeostasis generates oxidative stress through ROS accumulation and mitochondrial dysfunction, contributing to EC dysfunction. This mechanism, recently identified as “cuproptosis,” involves EC dysfunction, dysregulated inflammation, and tissue damage, playing a role in cardiovascular disease pathogenesis, including myocardial injury, coronary artery disease, and atherosclerosis [35,36]. Oxidative stress initiation by Cu occurs through cyclic transitions of Cu ions between oxidation and reduction states, generating hydroxyl radicals that cause DNA damage and lipid peroxidation [35]. ROS induce mitochondrial dysfunction and reduce ATP synthesis by impairing mitochondrial electron transport, damaging the inner mitochondrial membrane, and compromising mitochondrial DNA integrity [39] (Figure 5).

#### 3.1.4. Magnesium

Magnesium (Mg^2+^) is a macroelement whose daily requirement falls within the trace element range. It participates in cardiovascular function by influencing blood pressure through vasodilation. Mg^2+^ acts on ECs, stimulating NO and prostacyclin (PGI_2_) release and inhibiting L-type calcium channels.

Intracellular Ca^2+^ reduction in myocytes and the consequent weakening of cell contraction occurs not only via L-type calcium channel blockade but also through additional mechanisms. Therefore, the smooth muscle contraction process must be discussed. Contraction is generated by Ca^2+^ influx into smooth muscle cytosol via L-type calcium channels, followed by diacylglycerol (DG) and inositol 1,4,5-trisphosphate (IP_3_) synthesis and phospholipase C stimulation in response to intracellular Ca^2+^. The sarcoplasmic reticulum releases Ca^2+^ through IP_3_ receptor activation. Myosin light chain kinase (MLCK) is activated by calmodulin binding to cytosolic Ca^2+^, inducing cell contraction via myosin–actin interaction through myosin light chain phosphorylation [40]. Intracellular Mg^2+^ attenuates myocyte contraction at multiple steps. It has been shown to block L-type calcium channels, reducing Ca^2+^ influx by direct binding [40,41]. In cardiac muscle, Mg^2+^ inhibits Ca^2+^ release from the sarcoplasmic reticulum, which is triggered by Ca^2+^ itself [40,42]. After Ca^2+^ release from intracellular stores, elevated Ca^2+^ levels are prolonged under low-Mg^2+^ conditions, as Mg^2+^ facilitates Ca^2+^ reuptake into stores via sarcoplasmic reticulum Ca^2+^-ATPase activation [40].

Mg^2+^ may also influence the calcium-sensing receptor (CaSR), present in the parathyroids and expressed by vascular smooth muscle cells. Mg^2+^ can mimic calcimimetics, which inhibit vascular smooth muscle cell calcification via CaSR activation. This reduces parathyroid hormone (PTH) secretion, which regulates calcium homeostasis by increasing CaSR expression. Hemodialysis patients with higher serum Mg^2+^ show lower PTH levels. However, Mg^2+^ effects on vascular smooth muscle cell CaSR remain under investigation [42].

Another factor affecting vasodilation is PGI_2_, whose release in blood vessels is regulated by Mg^2+^ [40,41,43,44]. Extracellular Mg^2+^ increases PGI_2_ secretion in cultured ECs and smooth muscle cells. MgSO_4_ infusion has been shown to lower blood pressure, concomitantly increasing urinary excretion of 6-keto-prostaglandin F1α (6-keto-PGF1α), a stable PGI_2_ metabolite. The blood pressure–lowering effect of PGI_2_ is counteracted by cyclooxygenase inhibition, which prevents PGI_2_ synthesis and renal blood flow increase. Mg^2+^ may also affect cyclooxygenase activity via Ca^2+^ influx, as Mg^2+^-stimulated PGI_2_ release is inhibited by the calcium channel blocker nifedipine.

Mg^2+^ also modulates vascular tone by enhancing NO synthesis via eNOS upregulation in high-Mg^2+^ conditions [40]. Mg^2+^ deficiency has been associated with EC inflammation, oxidative stress, and intracellular lipid accumulation [41].

Mg^2+^ regulates membrane permeability for Na^+^ and K^+^. Fluctuations in Mg^2+^ levels may increase intracellular Na^+^ and Ca^2+^ and decrease K^+^ [42].

Mg^2+^ is essential for the activity and proper folding of hyaluronan synthase, which maintains extracellular matrix (ECM) structural integrity. The ECM provides vascular scaffolding and supports cellular processes and survival. Mg^2+^ modulates vascular cells via integrin transmembrane receptor binding, initiating intracellular signaling cascades. Integrin adhesion depends on metal ion availability, and Mg^2+^ concentration is critical for integrin–ligand interactions. Smooth muscle cell adhesion to the basement membrane is enhanced by fibronectin binding to the α5β1 integrin in vascular tissue. Higher Mg^2+^ levels increase α5β1–ligand stability, and Mg^2+^ is required for ligand binding site exposure [41].

Other mechanisms by which Mg^2+^ regulates blood pressure include: (1) modulating renal K^+^ handling, affecting systemic K^+^ homeostasis and indirectly blood pressure; (2) regulating aldosterone secretion from the adrenal cortex; (3) influencing catecholamine release from sympathetic nerve endings via N-type calcium channels and N-methyl-D-aspartate (NMDA) receptors; (4) modulating hypertensive factors in inflammation within macrophages and dendritic cells by stimulating the pyrin domain–containing NLR family member 3 (NLRP3) inflammasome and activating isolevuglandin (IsoLG) synthesis [40]. (Figure 6).

### 3.2. Heavy Metals

Monitoring and risk assessment of heavy metal exposure play a critical role in identifying vulnerable populations and evaluating the impact of environmental contaminants. Factors such as sex, age, lifestyle disorders, and socio-economic status must be considered. Exposure to heavy metals contributes to shortened life expectancy and an increased risk of cardiovascular diseases. This occurs through dietary intake and environmental contamination, including air pollution [45,46]. Suspended particulate matter may contain, among others, Pb and Cd. Particulate matter with a diameter of 0.1 µm represents ultrafine particles, which have high toxic potential for the cardiovascular system due to their ability to penetrate the alveolar–capillary barrier. This is facilitated by the increased number of particles, high solubility, elevated surface area-to-mass ratio, and reactive surface. A correlation has been demonstrated between heart failure and coronary artery disease risk and low-level Cd exposure, showing a linear dose–response relationship. Risk ratios for overall cardiovascular disease with 1 µg/L in whole blood and 0.5 µg/g creatinine in urine were 2.58 and 2.79, respectively.

Diet is a major route of heavy metal exposure. One study observed a significant association between Cd intake from vegetables and rice and its urinary excretion. Heavy metal absorption can be influenced by dietary factors, and essential minerals may increase absorption [45,47].

Increased exposure to Cd, Pb, and Hg has been noted in areas with high industrial activity or contaminated water sources. Children and the elderly are particularly susceptible [48].

Mercury (Hg), lead (Pb), and cadmium (Cd) are classified as non-essential heavy metals, meaning they have no known biological functions. They pose serious cardiovascular health risks, as even low doses can cause damage [49]. Their toxicity depends on bioaccumulation, dose, chemical properties, exposure duration, and route of entry. These metals disrupt antioxidant defenses and signaling pathways, affecting cellular processes such as metabolism, proliferation, survival, and apoptosis. They evade regulatory processes including homeostasis, transport, cellular signaling, compartmentalization, and antioxidant systems, partly through redox reactions [50,51].

A significant association has been demonstrated between elevated Hg, Pb, and Cd levels, high-sensitivity C-reactive protein (hs-CRP) levels, and ten-year cardiovascular disease risk [52]. Heavy metal exposure promotes hypertension by impairing NO signaling, inducing oxidative stress, disrupting Ca^2+^ signaling in vascular smooth muscle, causing renal damage, and altering the renin–angiotensin system. Cardiovascular dysfunction is linked to mitochondrial impairment in cardiac tissue [53].

High Hg exposure induces vasoconstriction, whereas low exposure promotes vasorelaxation. Vasoconstriction is associated with endothelial functional and compositional changes, while vasorelaxation is endothelium-dependent. Under low-Hg conditions, NO increases without affecting endothelium-derived hyperpolarizing factor (EDHF) or PGI_2_. Oxidative stress occurs under both high and low Hg exposures, leading to vessel constriction via NO interaction with superoxide anions and peroxynitrite (ONOO^−^) formation. High Hg reduces NO synthesis by decreasing eNOS activity, while EDHF levels remain constant or elevated. The endothelium can also release the potent vasoconstrictor endothelin-1 (ET-1), which exerts mitogenic and pro-inflammatory effects contributing to cardiovascular disease pathogenesis [54].

Exposure of human EA.hy926 ECs to methylmercury (MeHg) reduced NO synthesis, while other studies showed increased malondialdehyde (MDA) levels following exposure to MeHg and HgCl_2_. MDA is a lipid peroxide generated during lipid peroxidation and causes cellular damage [55]. Even very low Hg doses increase the phosphorylated eNOS-to-eNOS ratio, reducing vascular reactivity, suggesting toxic endothelial effects at low concentrations due to intracellular accumulation [56]. Studies demonstrate that Hg^2+^ and MeHg induce EC apoptosis and necrosis, contributing to atherogenesis [57]. MeHg also promotes monocyte adhesion to HMEC-1 ECs and upregulates pro-inflammatory cytokines and adhesion molecules, including chemotactic cytokines and intercellular adhesion molecule-1 (ICAM-1), significantly activating NF-κB [58].

Hg induces inflammation via TNF-α, pro-inflammatory cytokines, IL-2, IL-6, and IL-10, and modulates cellular signaling pathways. High Hg concentrations positively correlate with CRP levels [49].

Hg-generated oxidative stress, through increased pro-oxidant production or decreased antioxidant activity, affects vulnerable ECs, causing inflammation and subsequent dysfunction [59]. Hg may damage cardiac tissue via oxidative stress induction, mitochondrial dysfunction, and sulfhydryl group depletion [60]. Chronic Hg exposure has been shown to contribute to cardiovascular disease development via endothelial dysfunction, inflammation, and oxidative stress [61].

Pb contributes to hypertension by increasing vascular tone through multiple cellular mechanisms, including oxidative stress, inflammation, decreased NO and EDHF levels, disrupted Ca^2+^ transport, elevated endothelin levels, and reduced vasoactive hormones. Pb may enhance ROS production and oxidative stress via Fenton and Haber–Weiss reactions. It also impairs Na^+^/K^+^ pump function, signaling pathways, and myofibril phosphorylation, affecting cardiac contractility and conduction. Studies show Pb increases eNOS expression while reducing soluble guanylyl cyclase (sGC) expression, suggesting effects on endothelial relaxation and vascular remodeling [49,62,63]. Pb selectively binds sulfur-containing enzymes and antioxidants, generating ROS and depleting cellular antioxidant reserves [62]. Furthermore, Pb affects Zn- and Ca-dependent functions, primarily antioxidant activity, by reducing SOD activity and inhibiting glutathione synthesis [64]. Chronic Pb exposure decreases antioxidant defenses by downregulating SOD, GPx, and catalase expression [65].

Low-dose Pb exposure impairs iron utilization and promotes accumulation of unstable iron in mitochondria, disrupting NO metabolism and causing endothelial dysfunction. Pb exerts membrane toxicity in cardiomyocytes [66]. Exposure of ECs to Pb reduces their protective capacity by disturbing cellular processes and morphology [6,67]. A link has also been found between Pb and endothelial nitric oxide synthase gene (NOS3), correlating with susceptibility to cardiovascular disease. The NOS3 Glu298Asp polymorphism in exon 7 (G → T at nucleotide position 298) is involved [68].

Studies analyzing IL-8 production in human umbilical vein ECs exposed to Pb showed that Pb affects Nrf2 signaling [69]. Pb also increases soluble adhesion molecules in blood, potentially causing endothelial damage [47]. Population studies in adolescents and young adults revealed positive correlations between urinary Pb levels and endothelial and platelet microparticles, partly explaining Pb-induced cardiotoxicity. Endothelial/platelet activation and apoptosis due to Pb accumulation may contribute to atherosclerosis pathogenesis [70].

Cd contributes to hypertension and atherosclerosis by impairing EC and smooth muscle function. Cd exposure increases asymmetric dimethylarginine (ADMA), an endogenous eNOS inhibitor, and decreases L-arginine, which promotes NO production. Other studies demonstrate that Cd disrupts lipid metabolism in human microvascular ECs (HMEC-1), causing triglyceride breakdown and free fatty acid accumulation, generating ROS, cytotoxicity, reduced ATP, and altered mitochondrial membrane potential, resulting in endothelial dysfunction and cell death. Cd exposure also inhibits EC proliferation and induces apoptosis. Acute Cd exposure increases vasoconstrictor activity via NADPH oxidase–mediated ROS production, reducing NO bioavailability and promoting endothelial oxidative stress, leading to vascular damage [54,71,72].

Cd contributes to hypertension by suppressing acetylcholine-induced vasodilation and inhibiting endothelial NO synthase [73]. Cd promotes atherosclerosis and vascular inflammation via two mechanisms: (1) it penetrates vessel walls via monocytes through transporters and ion channels; these monocytes and macrophages eventually cause foam cell necrosis, releasing cytokines and Cd, resulting in endothelial damage and local inflammation; (2) Cd disrupts endothelial integrity by inducing EC death, forming gaps that allow Cd to reach the medial layer. Cd is retained in smooth muscle cells, where low concentrations disrupt calcium flow and ionic homeostasis, promote smooth muscle proliferation, and exert cytotoxic effects. These processes contribute to an atherogenic state via lipid accumulation and lipid profile modification. Cd-induced cell death impairs endothelial integrity, promoting atherosclerosis and vascular inflammation [74,75]. Cd exposure also induces hypertension via increased peripheral resistance through RAS activation and endothelial dysfunction [76]. Low-dose Cd exposure has been associated with vascular aging, as evidenced by elevated plasma von Willebrand factor (vWF), a glycoprotein involved in coagulation. vWF levels rise with age, contributing to plaque formation, inflammation, thrombosis, and vascular smooth muscle cell proliferation [77].

It is also worth noting that the human body is exposed to heavy metals through three main routes: the gastrointestinal route, the respiratory route, and penetration through the skin. The health risk is determined by the route of exposure, the duration of exposure, and the dose. Whether metal poisoning is chronic or acute is most often influenced by the magnitude of the dose. Factors such as the oxidation/reduction capacity of the metal ion, its valence state, ionic radius, ability to interact with the ligand environment, and lipophilicity/hydrophilicity inform the degree of toxicity of a given metal. Common features of these processes include dysregulation of antioxidant mechanisms, oxidative stress generated by reactive oxygen species (ROS), and enzyme inactivation.

Below are several examples of dose–exposure relationships for heavy metals. The effect of Cd on the activity of the sodium–potassium pump can be reversed at low/mild cellular intoxication after an initial decrease in ATPase activity. The toxic effect predominates when the exposure dose is very high. In turn, studies of exposure to Hg at high doses have shown an association with cytotoxicity. However, exposure to low doses of this metal caused increased DNA damage, among other effects, through blockage of DNA repair mechanisms and inhibition of apoptosis. Another study demonstrated a dose-dependent relationship between Pb exposure and the severity of cardiovascular disease. Low blood Pb levels (<5 μg/dL) observed in the general population were associated with hypertension and mortality due to peripheral arterial disease, stroke, and coronary heart disease [50].

## 4. Consequences of Endothelial Dysfunction

### 4.1. Cardiovascular Disease Development

Endothelial dysfunction, resulting from the predominance of pro-inflammatory factors and oxidative stress over protective mechanisms, constitutes a key mechanism in the pathogenesis of cardiovascular diseases. One of the most critical aspects is that endothelial dysfunction contributes to the initiation and progression of atherosclerotic plaque formation, making it considered the earliest stage of most cardiovascular disorders [78]. Vascular homeostasis relies on the balance between vasodilatory mediators, such as nitric oxide (NO) and prostacyclin, and vasoconstrictive factors, including endothelin-1. Chronic exposure to reactive oxygen species (ROS), exacerbated by the presence of heavy metals such as cadmium, lead, or mercury, leads to NO inactivation, reducing vascular relaxation capacity and promoting inflammatory processes [21,79].

In this context, vascular endothelial inflammation is recognized as a significant factor contributing to the development of numerous cardiovascular-related conditions, including hypertension, atherosclerosis, aging, stroke, heart disease, diabetes, obesity, venous thrombosis, and pathological vascular wall thickening [80].

#### 4.1.1. Atherosclerosis

Oxidative imbalance leads to decreased NO bioavailability and endothelial dysfunction, favoring the formation of atherosclerotic lesions through enhanced leukocyte adhesion, inflammation, and cellular structural damage [81]. In damaged endothelium, excessive production of pro-inflammatory cytokines and chemokines activates complex cellular cascades [21].

Studies show that selenium-containing compounds, such as SeTal or the GPX-1 mimetic ebselen, reduce oxidative stress, increase NO bioavailability, and inhibit pro-inflammatory signaling pathways. Selenoproteins, including GPX-1 and SelS, modulate Akt/eNOS and NF-κB signaling, decrease ROS production, reduce adhesion molecule expression, and limit apoptosis in endothelial cells. This indicates that selenium-dependent enzymes play a crucial role in endothelial protection and atherosclerosis prevention [21].

Zinc-dependent transcription factors, including KLF2, KLF4, and KLF11, exhibit strong anti-inflammatory and anti-atherosclerotic effects, protecting endothelium from pro-inflammatory signaling activation. Zinc also supports PPAR-α and PPAR-γ receptor activity, which inhibits NF-κB activation and reduces vascular inflammation, contributing to the slowing of atherosclerosis progression [29].

A large population study demonstrated that individuals in the highest quartile of blood lead concentration had approximately a 35% higher risk of developing atherosclerotic changes compared with those in the lowest quartile. This effect was particularly pronounced in postmenopausal women, who exhibited a 72% increased risk, highlighting the role of lead as an accelerator of carotid artery changes, especially in vulnerable groups [82].

#### 4.1.2. Hypertension

The relationship between developing hypertension and endothelial dysfunction involves regulatory and structural disturbances at multiple levels. Damaged endothelial cells promote altered smooth muscle vascular reactivity while simultaneously inducing prothrombotic and pro-inflammatory effects, exacerbating organ damage associated with hypertension [26]. Mercury’s role in hypertension pathophysiology may involve endothelial dysfunction, renin–angiotensin system (RAS) disturbance, and inflammation. These components are interrelated, leading to hypertension through a complex mechanism. Mercury stimulates the renin–angiotensin–aldosterone (RAA) system, increasing angiotensin-converting enzyme (ACE) and renin secretion, resulting in elevated angiotensin II, a potent vasoconstrictor. Angiotensin II stimulates ROS production in vascular smooth muscle cells. Additionally, mercury acts directly on endothelial cells, forming complexes with glutathione (GSH) and selenium, disrupting redox balance and impairing ROS clearance. Excess ROS activates the NF-κB transcription factor, increasing pro-inflammatory cytokine expression, including TNF-α and IL-6, exacerbating inflammation, and promoting atherosclerosis. ROS also inhibits NO and cyclic GMP (cGMP), leading to vascular constriction and increased resistance, fostering hypertension development. Studies also show that NO is directly reduced by HgCl_2_ and MeHg [55]. Wang et al. examined the correlation between lead, cadmium, mercury, and manganese levels and hypertension. Lead emerged as the dominant factor influencing hypertension development, similar to mercury, reducing NO bioavailability, impairing antioxidant systems, and increasing ROS production [62]. The SWAN study confirmed this correlation, showing that urinary arsenic, mercury, and lead levels were associated with faster increases in systolic and diastolic blood pressure in menopausal women. Cadmium was linked to accelerated systolic pressure rise in never-smoking women. Over 17 years, mean systolic and diastolic pressures increased by 15.8 mmHg and 12.6 mmHg, respectively, in the highest versus lowest arsenic tertiles [83].

#### 4.1.3. Obesity

In obesity, adipose tissue secretes numerous hormones and cytokines that disrupt metabolism and promote endothelial dysfunction, contributing to atherosclerosis. Inflammatory activation and MCP-1 increase adhesion molecule expression, enhancing leukocyte adhesion and migration and elevating the risk of atherothrombotic events. Individuals with obesity and metabolic syndrome exhibit elevated endothelial damage markers, such as VEGF-1, increased ROS production, higher PAI-1 promoting thrombosis, hyperuricemia, elevated triglycerides, and easily oxidized LDL [84].

Zinc supplementation can protect endothelium through multiple mechanisms, including reducing NF-κB–dependent inflammatory mediator expression, enhancing eNOS expression and NO production, activating PPARs, and inhibiting TNF-α–induced apoptosis [85].

Animal studies demonstrate the role of zinc in regulating cardiovascular parameters, including the reduction in adipose tissue mass. Obese animals showed increased oxidative stress, elevated glucose, lipid levels, leptin, and blood pressure, promoting endothelial dysfunction and cardiac damage. Zinc deficiency exacerbated oxidative stress. Administration of zinc oxide nanoparticles (ZnONP) mitigated these changes, reducing cardiomyocyte damage, vascular wall hypertrophy, periaortic fat deposition, and inflammatory markers such as TNF-α, CRP, IL-6, resistin, and MCP-1 [86].

Similarly, zinc impacts obesity development via endothelial dysfunction mechanisms. Obese children exhibit reduced selenium and selenoprotein P levels. Obesity increases micronutrient demand, and selenium deficiency weakens antioxidant protection and enhances inflammation [87]. NHANES 2003–2014 data indicated correlations between heavy metal exposure and obesity indices, including waist circumference, BMI, skinfold thickness, and total body fat [88].

#### 4.1.4. Diabetes

Diabetes is largely associated with vascular changes, though precise mechanisms remain incompletely understood. Current reviews suggest that hyperglycemia increases oxidative stress and endothelial inflammation, reducing vasodilatory capacity and promoting atherosclerosis [89]. Diabetes is associated with chronic microvascular and macrovascular complications. EPIC cohort studies demonstrated sex-specific differences: higher zinc levels in women were linked to increased risk of general and microvascular complications, whereas in men, higher zinc was associated mainly with macrovascular complications. A low copper-to-zinc ratio decreased microvascular complication risk in women but increased it in men [90].

Even relatively low arsenic exposure may be linked to cardiovascular risk factors, including diabetes. NHANES 2003–2004 data showed higher urinary arsenic (median 7.1 μg/L) was associated with increased type 2 diabetes prevalence [91]. Another Mexican cross-sectional study linked water arsenic (25.5–47.9 μg/L) and total urinary arsenic (<55.8 μg/L) with diabetes and dyslipidemia [92]. Peining Liu et al. demonstrated that elevated glucose increases ROS production and decreases GTPCH1 and BH4 in endothelial cells. GTP cyclohydrolase 1 (GTPCH1) is essential for tetrahydrobiopterin (BH4) synthesis, a critical cofactor for endothelial NO synthase (eNOS). In this state, eNOS produces ROS instead of NO, aggravating oxidative stress and endothelial dysfunction. Hyperglycemia also promotes zinc release from GTPCH1, impairing its enzymatic function. Zinc supplementation can reverse these changes by stabilizing GTPCH1, restoring BH4 levels, and reducing oxidative stress [93]. Mitochondrial dysfunction, impaired endothelial repair, and increased permeability are additional mechanisms explored [21].

#### 4.1.5. Hyperlipidemia

Yangping Zhuang et al. showed that higher blood levels of lead, cadmium, and selenium are associated with increased hyperlipidemia risk. A major mechanism is enhanced oxidative stress, as these metals bind to protein –SH groups, generating ROS, which deplete glutathione, the body’s natural antioxidant. Rising oxidative stress triggers inflammatory responses, a key factor in lipid disorders like hyperlipidemia. Additionally, heavy metals affect lipid metabolism in the liver and adipose tissue by reducing antioxidant enzyme activity (e.g., superoxide dismutase, catalase), promoting lipid peroxides, and disrupting redox balance, which may damage hepatocytes and impair lipid production [94].

Heavy metal exposure (e.g., lead, cadmium, mercury) contributes to endothelial dysfunction via NO inactivation, oxidative stress, inflammation, vascular smooth muscle damage, and mitochondrial and immune disturbances. Exposure also reduces systemic antioxidant capacity and disrupts lipid profiles, decreasing HDL and increasing LDL, total cholesterol, triglycerides, and CRP levels. Vasoconstrictive prostaglandins are elevated, further impairing vascular function. Cardiologically, heavy metals may cause ECG changes, catecholamine-induced arrhythmia susceptibility, and structural myocardial damage, including fibrosis and inflammation. Mechanistically, these effects are linked to catechol-O-methyltransferase inhibition, leading to increased adrenaline, noradrenaline, and dopamine, contributing to hypertension [79]. Chronic low-level heavy metal exposure is associated with increased coronary artery disease risk, especially in older adults [95].

### 4.2. Preventive Strategies

Prevention of cardiovascular diseases is widely discussed and promoted by primary care physicians, with particular emphasis on lifestyle and health behaviors. These include, for example, a balanced diet, regular physical activity, and a reduction in substance use. Regarding deficiencies or excesses of trace elements, prevention of such states primarily falls within the scope of a balanced diet. A key element of prevention is maintaining adequate levels of trace elements in the body. Concentrations of these elements, dependent on various factors, can be regulated through appropriate mineral supplementation or dietary restriction.

Given the crucial role of the endothelium and the involvement of trace elements in its regulation, it is worth reviewing studies highlighting the value of supplementation for reducing cardiovascular risk. In a large cross-sectional study including nearly 40,000 adults, the impact of 11 trace elements on cardiovascular risk was evaluated, including zinc, copper, and selenium. The study concluded that sufficiently high concentrations of antioxidant trace elements may reduce cardiovascular risk, with selenium showing the greatest effect. However, analyses varied depending on whether the combined or individual effects of each element were considered [96]. Literature reports indicate that selenium supplementation at doses of 55 μg/day or 200 μg/day for 12 weeks exhibits antioxidant activity and reduces cardiometabolic risk [97,98]. Other studies show that magnesium supplementation (mean 400 mg/day) lowered triglyceride levels and increased HDL-C, whereas zinc (mean 30 mg/day) decreased total cholesterol and triglycerides [99]. According to the EFSA Panel on Nutrition, Novel Foods and Food Allergens (NDA), reference values for selenium and copper are 255 μg/day and 0.07 mg/kg body weight/day, respectively. Maintaining trace element concentrations within recommended ranges may play an important role in cardiovascular risk prevention and in supporting proper endothelial function.

Dietary education promoting a balanced diet rich in vegetables, nuts, whole grains, and seeds may provide the greatest benefit. The permissible values of heavy metals in water and soil, as recommended by WHO and UPEA, are: lead 0.01/0.005 mg/L, mercury 0.001/0.002 mg/L, and cadmium 0.003/0.005 mg/L. Research on soil decontamination techniques, such as phytoremediation, chemical, or physical methods, may contribute to reducing exposure to these metals [100]. Heavy metals like cadmium are present in tobacco smoke. Studies have shown that blood cadmium levels increase in smokers, correlating with higher risk of cardiovascular events. Smokers also exhibit shorter life expectancy and higher incidence of cardiovascular disease, with cadmium accounting for up to half of this risk. Therefore, smoking cessation is a key strategy for reducing exposure to toxic metals and preventing heart disease [101]. In terms of prevention, reducing exposure to heavy metals through elimination of tobacco smoke (a source of cadmium and arsenic), control of water and food quality, environmental actions including soil remediation and purification systems, and a reduction in industrial pollutants is critical. On the other hand, rather than relying solely on the removal of heavy metals, for example, through chelation therapy, which may disrupt the balance of essential trace elements, a more physiological preventive strategy involves strengthening endogenous detoxification and defense mechanisms at the level of endothelial cells. In selected cases, targeted supplementation, particularly with zinc, selenium, or magnesium, may be justified at doses adjusted to deficiency levels. In contrast, synergistic supplementation of these trace elements may exert a greater effect by targeting the same pathways at different regulatory levels. Growing evidence suggests that the gut microbiome plays an important role in regulating the bioavailability of heavy metals and the absorption of essential trace elements. Microbiome dysbiosis may enhance the intestinal uptake of cadmium, lead, and mercury, thereby exacerbating oxidative stress and endothelial dysfunction. In this context, targeted modulation of the gut microbiome through dietary or prebiotic interventions may represent an innovative strategy for cardiovascular disease prevention, acting as a first-line barrier against environmental heavy metal exposure [102,103]. Circadian rhythms play a critical role in regulating endothelial function, oxidative stress, and the activity of signaling pathways involved in vascular homeostasis. Increasing attention has been directed toward the concept that the efficacy of nutritional and supplementation interventions may depend not only on the administered dose but also on the timing of intake. Chronobiology-based optimization of trace element supplementation may allow for more effective modulation of endothelial protective mechanisms [104].

## 5. Conclusions

This review analyzed the impact of trace elements and heavy metals on endothelial function and their relationship to cardiovascular risk in the context of dynamic environmental changes. The endothelium, as an active biological structure, plays a central role in maintaining vascular homeostasis, regulating vascular tone, permeability, leukocyte adhesion, and coagulation. Any disruption of its function, termed endothelial dysfunction, leads to a predominance of pro-inflammatory and pro-oxidative mechanisms, resulting in the development of cardiovascular conditions such as atherosclerosis, hypertension, diabetes, obesity, and hyperlipidemia.

Trace elements, including zinc, copper, magnesium, and selenium, exhibit significant protective effects on endothelial cells. They act as cofactors for antioxidant enzymes (e.g., superoxide dismutase, catalase, glutathione peroxidase), supporting neutralization of reactive oxygen species (ROS) and a reduction in oxidative stress. Deficiencies in trace elements can disrupt these mechanisms, exacerbate oxidative stress, and lead to vascular dysfunction.

In contrast, heavy metals such as lead, cadmium, mercury, and arsenic adversely affect the endothelium. By binding to protein thiol groups and depleting antioxidant reserves, they induce oxidative stress and chronic inflammation, reduce NO bioavailability, and impair mitochondrial function. These relationships are confirmed by numerous epidemiological studies, including NHANES, SWAN, and studies conducted in Mexico and Asia [62,76,88,91,92]. The evidence clearly indicates that heavy metal effects on endothelial function are multifactorial and highly toxic, even at low-level chronic exposure, which may remain clinically silent for many years. Thus, this issue represents not only an individual health concern but also a public health challenge.

The innovation of this review lies in its comprehensive approach to the interactions between heavy metals, trace elements, and vascular endothelium, with reference to increasing environmental threats. It integrates knowledge from molecular biology, environmental toxicology, and preventive cardiology, emphasizing the importance of an interdisciplinary approach to cardiovascular risk assessment.

Future research directions should include several key areas. First, the long-term impact of low-level heavy metal exposure on endothelial function should be evaluated, considering the roles of epigenetics and the microbiome. Clinical studies assessing the efficacy of trace element supplementation in reversing early endothelial changes are also essential. Another goal should be identifying biomarkers of environmental toxicity at the cellular level and strategies to eliminate exposure from daily life.

In summary, proper endothelial function is a major determinant of cardiovascular health. Both trace element deficiencies and exposure to toxic heavy metals significantly affect its function. Environmental and nutritional prevention, along with research on the mechanisms of action of these substances, can form the foundation for effective strategies to reduce cardiovascular morbidity in the 21st century.

## Figures and Tables

**Figure 1 cimb-48-00041-f001:**
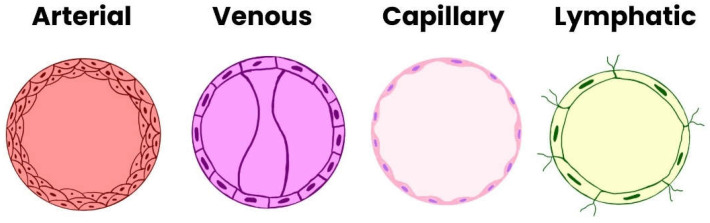
Types of Endothelial Cells.

**Figure 2 cimb-48-00041-f002:**
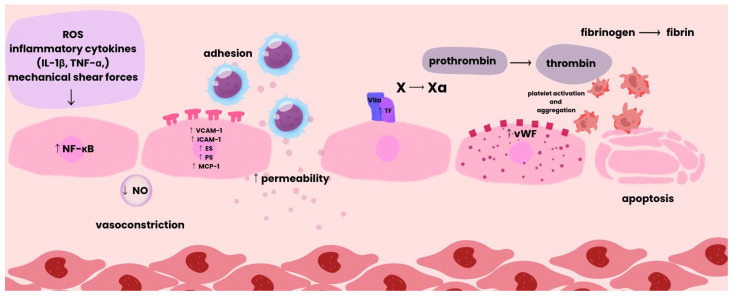
Pathological Sequence of Vascular Damage.

**Figure 3 cimb-48-00041-f003:**
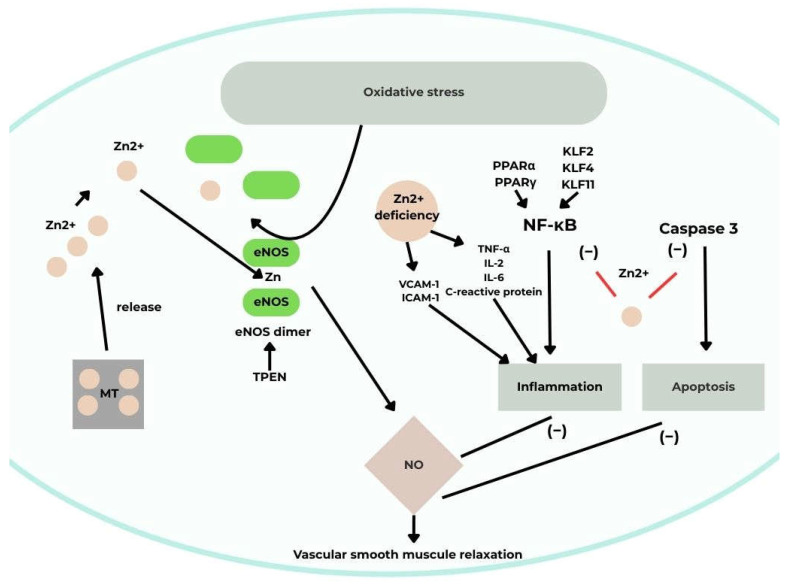
A schematic illustrating the effect of Zn on vascular endothelial function [30]. Acronyms: MT, metallothionein; eNOS, endothelial nitric oxide synthase; NO, nitric oxide; TNF-α tumor necrosis factor-alpha; NF-κB, nuclear factor-κB; TPEN, N,N,N,N-tetrakis(2-pyridylmethyl)-ethylenediamine; IL-2, interleukin 2; IL-6; interleukin 6; VCAM-1, vascular cell adhesion molecule-1; ICAM-1, intercellular adhesion molecules; PPARα, peroxisome proliferator-activated receptor alpha; PPARγ, peroxisome proliferator-activated receptor gamma; KLF2, Krüppel-like factor 2; KLF4, Krüppel-like factor 4, KLF11, Kruppel-like factor 11; Zn^2+^, zinc (II) ion.

**Figure 4 cimb-48-00041-f004:**
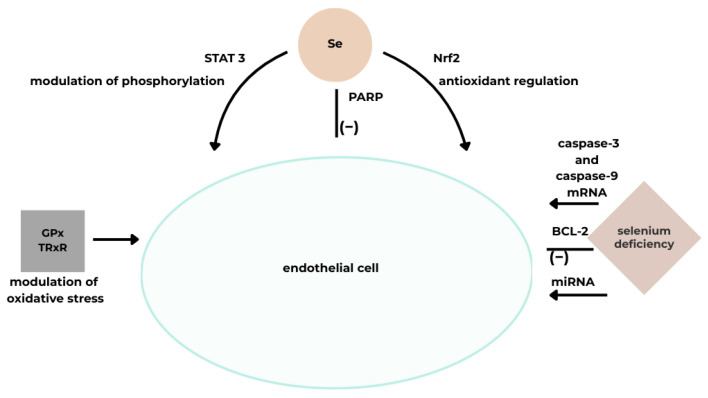
A schematic illustrating the effect of Se on vascular endothelial function [5]. Acronyms: GPx, glutathione peroxidase; TRxR, thioredoxin reductase; Nrf2, nuclear factor-erythroid 2-related factor; BCL-2, B-cell CLL/lymphoma 2; miRNA, microRNA; mRNA stands for messenger ribonucleic acid; PARP, Poly(ADP-ribose) polymerase; STAT 3, stands for Signal Transducer and Activator of Transcription 3, Se, selenium.

**Figure 5 cimb-48-00041-f005:**
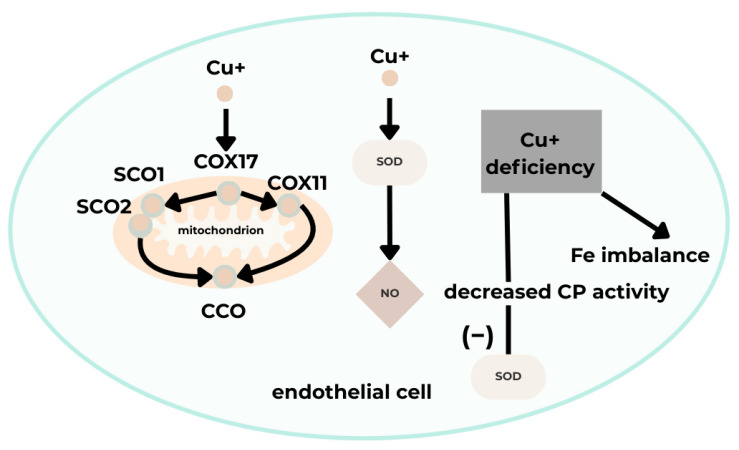
A schematic illustrating the effect of Cu on vascular endothelial function [35]. Acronyms: COX11, cytochrome c oxidase copper chaperone 11; COX17, cytochrome C oxidase copper chaperone 17; SCO1, synthesis of cytochrome C oxidase 1; SCO2, synthesis of cytochrome C oxidase 2; CCO, cytochrome C oxidase; CP, ceruloplasmin; SOD, superoxide dismutase; NO, nitric oxide; Cu+, copper (I) ion; Fe, iron.

**Figure 6 cimb-48-00041-f006:**
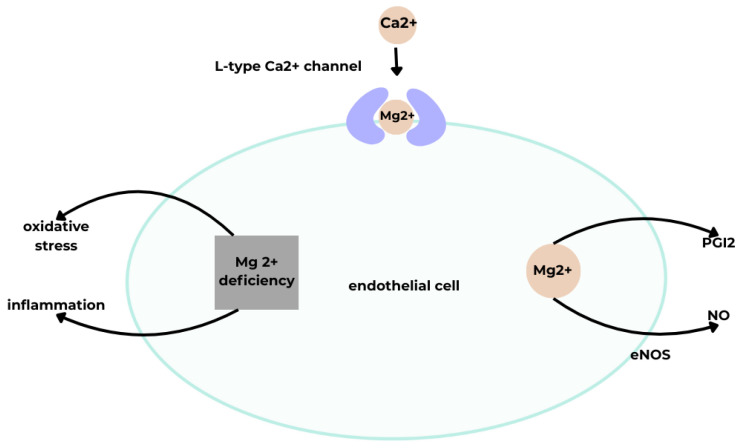
A schematic illustrating the effect of Mg on vascular endothelial function [40]. Acronyms: eNOS, endothelial nitric oxide synthase; NO, nitric oxide; PGI2, prostacyclin; Mg^2+^, magnesium (II) ion; Ca^2+^, calcium (II) ion.

**Table 1 cimb-48-00041-t001:** Types of Endothelial Cells [14].

Type of Endothelial Cell	Structural Characteristics	Primary Function
Arterial	Typically flat, elliptical, or elongated and narrow; do not form valves.	Transport of oxygen and nutrients to peripheral tissues.
Venous	Short and wide; form intraluminal valves.	Transport of deoxygenated blood from tissues to the heart.
Capillary	Thin structure with numerous caveolae.	Exchange of oxygen, nutrients, and metabolic products within the circulatory system.
Lymphatic	Thin basement membrane with anchoring filaments.	Mediation of immune responses and maintenance of interstitial fluid balance.

**Table 2 cimb-48-00041-t002:** Endothelial vasoactive mediators [26].

Mediator	Mechanism of Action	Additional Effects
Nitric Oxide (NO)	Produced by eNOS from L-arginine; activates guanylate cyclase → ↓ intracellular Ca^2+^, ↑ K^+^ → smooth muscle relaxation	Antithrombotic, anti-inflammatory, antiproliferative, anti-atherosclerotic
EDHF (Endothelium-Derived Hyperpolarizing Factor)	Opens Ca^2+^-activated K^+^ channels → hyperpolarization of vascular smooth muscle membrane → vasodilation	Particularly active in microcirculation; role in hypertension not fully elucidated
Prostacyclin (PGI_2_)	Activates adenylate cyclase → ↑ cAMP → smooth muscle relaxation	Inhibits platelet aggregation; acts synergistically with NO
Angiotensin 1–7	Formed from Ang II by ACE2; activates MAS receptor → ↑ NO, ↓ ROS	Endothelial protective effects; antihypertensive properties
Angiotensin II (via AT2 receptor)	Binds AT2 receptor → signaling pathways leading to vasodilation	Opposes AT1 receptor-mediated effects; weaker binding affinity
Endothelin-1 (ET-1)	Activates ET_A_ and ET_B2_ receptors → ↑ Ca^2+^, myosin light chain phosphorylation → smooth muscle contraction	Mitogenic, pro-inflammatory, and vascular remodeling; at high concentrations, vasoconstrictive effects predominate
Angiotensin II (AT1 receptor)	Activates the AT1 receptor → ↑ ROS, ET-1, growth factors, adhesion molecules → vasoconstriction, vascular remodeling	Promotes oxidative stress, inflammation, and endothelial dysfunction; a key mechanism in hypertension
ACE	Converts Ang I → Ang II → activation of the AT1 receptor	Indirect mediator of vasoconstriction; particularly active in the pulmonary endothelium
ROS	Scavenge NO → form toxic peroxynitrite; increase oxidative stress	Promote inflammation, vascular remodeling, endothelial damage, and vessel narrowing

## Data Availability

No new data were created or analyzed in this study. Data sharing is not applicable to this article.

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
