# Peer review of "Effects of Micronutrients and Heavy Metals on Endothelial Function and Cardiovascular Risk in the Face of Environmental Changes"

_cimb, 2025, doi:10.3390/cimb48010041_

Round 1
Reviewer 1 Report
Comments and Suggestions for Authors
This is a comprehensive review on the effects of micronutrients and heavy metals on endothelial function and cardiovascular risk in the face of environmental changes. The review is timely with an emphasis on essential trace elements, such as zinc, selenium, copper, and magnesium, that support endothelial function through antioxidant defense, nitric oxide regulation, and anti-inflammatory signaling. Conversely, exposure to heavy metals including cadmium, lead, mercury, and arsenic that triggers oxidative stress, disrupts nitric oxide bioavailability, and promotes endothelial dysfunction, accelerating the pathogenesis of many diseases.
The subheadings and discussions were clear.
The manuscript will be enhanced by a Figure showing the different types of endothelial cells
Author Response
Comments 1: This is a comprehensive review on the effects of micronutrients and heavy metals on
endothelial function and cardiovascular risk in the face of environmental changes. The review is
timely with an emphasis on essential trace elements, such as zinc, selenium, copper, and
magnesium, that support endothelial function through antioxidant defense, nitric oxide regulation, and
anti-inflammatory signaling. Conversely, exposure to heavy metals including cadmium, lead, mercury,
and arsenic that triggers oxidative stress, disrupts nitric oxide bioavailability, and promotes endothelial
dysfunction, accelerating the pathogenesis of many diseases.
The subheadings and discussions were clear. The manuscript will be enhanced by a Figure showing the different types of endothelial cells.
Response 1: We sincerely thank the reviewer for their thorough evaluation and constructive
feedback. We appreciate the recognition of the comprehensiveness and timeliness of our review,
particularly regarding the roles of essential trace elements and heavy metals in endothelial function
and cardiovascular risk.
Regarding the suggestion to include a figure showing the different types of endothelial cells, we agree
that this addition would enhance the clarity and visual appeal of the manuscript.
Thank you again for your valuable comments and for helping us improve the quality of our work.
Reviewer 2 Report
Comments and Suggestions for Authors
This review by Agata Doligalska-Dolina et.al. discusses the specific role of heavy metals in endothelial function and cardiovascular risk in the face of environmental changes. The manuscript is well-organized, clearly written, and covers a broad range of relevant studies. However, it requires further improvements in content and clarity.
- Inclusion of a Schematic Mechanism Figure:
The textual description of mechanisms is detailed but could be more accessible. I recommend adding a summary figure that visually integrates the key pathways through which different metal impairs endothelial function. This would help readers, especially those new to the field, quickly grasp the central pathological processes and compare the actions of various metals.
- Suggestion on Dose-Response:
It would be valuable if the author could expand the discussion to address the potential for dose-dependent heterogeneity. Specifically, do different exposure levels (e.g., low chronic vs. high acute) of the same metal trigger distinct pathways?
Author Response
Comments 1: Inclusion of a Schematic Mechanism Figure:The textual description of mechanisms is
detailed but could be more accessible. I recommend adding a summary figure that visually integrates
the key pathways through which different metal impairs endothelial function. This would help readers,
especially those new to the field, quickly grasp the central pathological processes and compare the
actions of various metals.
Response 1: We sincerely thank the reviewer for the careful evaluation of our manuscript and for the
constructive suggestions that helped us improve its content and clarity.Regarding the
recommendation to include a schematic mechanism figure, we fully agree that a visual summary
would enhance the accessibility of the mechanistic discussion. In response, we have added new
schematic figures that integrates the key pathways through which essential trace elements, including
zinc, selenium, magnesium, and copper, influence endothelial function. The figure summarizes their
roles in antioxidant defense, nitric oxide bioavailability, inflammatory signaling, and maintenance of
endothelial integrity, allowing for easier comparison of their mechanisms of action and improved clarity
for readers new to the field.
Comments 2: Suggestion on Dose-Response:
It would be valuable if the author could expand the discussion to address the potential for
dose-dependent heterogeneity. Specifically, do different exposure levels (e.g., low chronic vs. high
acute) of the same metal trigger distinct pathways?
Response 2: Concerning the suggestion to address dose–response relationships, we appreciate this
important point. We have expanded the discussion to address potential dose-dependent
heterogeneity of essential trace elements, emphasizing that physiological versus excessive or
deficient levels of zinc, selenium, magnesium, and copper may differentially affect endothelial
pathways. Where available, we discuss how variations in exposure or intake levels can activate
distinct molecular mechanisms relevant to endothelial function.
We thank the reviewer again for these insightful comments, which have significantly strengthened the
manuscript.
Reviewer 3 Report
Comments and Suggestions for Authors
The article entitled “Metals” provides a solid description of the endothelium and its important biological functions at the introduction. The authors should pay attention to the quality of the figure they uploaded and perhaps improve the information they present with an image, as a single diagram is too ambiguous. Tables 1 and 2 are not cited in the text.
It would be helpful to better describe the environmental and physiological factors that contribute to endothelial dysfunction. Some acronyms are not explained.
The preventive strategies described are not particularly strong or novel enough to make for an interesting proposal.
The conclusions are very long and redundant; they should be shortened.
Author Response
Comments 1: The authors should pay attention to the quality of the figure they uploaded and
perhaps improve the information they present with an image, as a single diagram is too ambiguous.
Response 1: We sincerely thank the reviewer for the careful assessment of our manuscript
and for the constructive comments that helped us improve its quality.
Regarding the quality and clarity of the figures, we acknowledge the reviewer’s concern. The
previously included diagram has been revised and replaced with improved, higher-quality
schematic figures that provide clearer and more detailed visual information, reducing
ambiguity and better supporting the textual content.
Comments 2: Tables 1 and 2 are not cited in the text.
Response 2: We also apologize for the omission of citations to Tables 1 and 2 in the main
text. These tables are now explicitly cited and appropriately integrated into the relevant
sections of the manuscript.
Comments 3: It would be helpful to better describe the environmental and physiological factors that
contribute to endothelial dysfunction.
Response 3: In response to the suggestion to better describe environmental and
physiological factors contributing to endothelial dysfunction, we have expanded the
corresponding sections to include additional discussion on relevant environmental
exposures, nutritional status, oxidative stress, inflammation, and metabolic conditions.
Comments 4: Some acronyms are not explained.
Response 4: Furthermore, all acronyms are now defined at first mention to improve clarity
for the reader.
Comments 5: The preventive strategies described are not particularly strong or novel enough to
make for an interesting proposal.
Response 5: With regard to the preventive strategies, we have revised this section to
strengthen its scientific relevance and clarifying the potential role of micronutrient balance
and lifestyle-related factors in endothelial protection. While we acknowledge that this is a
review article, we aimed to improve the coherence and translational value of this section.
Comments 6: The conclusions are very long and redundant; they should be shortened.
Response 6: Finally, in accordance with the reviewer’s recommendation, the Conclusions
section has been substantially shortened and streamlined to eliminate redundancy and
improve focus.
We thank the reviewer again for these valuable comments, which have significantly
improved the clarity and overall quality of the manuscript.